# Adapting the coordination of eyes and head to differences in task and environment during fully-mobile visual exploration

**John M. Franchak◉\*, Brianna McGee, Gabrielle Blanch**

Department of Psychology, University of California, Riverside, Riverside, California, United States of America

\* john.franchak@ucr.edu

## Abstract

How are eyes and head adapted to meet the demands of visual exploration in different tasks and environments? In two studies, we measured the horizontal movements of the eyes (using mobile eye tracking in Studies 1 and 2) and the head (using inertial sensors in Study 2) while participants completed a walking task and a search and retrieval task in a large, outdoor environment. We found that the spread of visual exploration was greater while searching compared with walking, and this was primarily driven by increased movement of the head as opposed to the eyes. The contributions of the head to gaze shifts of different eccentricities was greater when searching compared to when walking. Findings are discussed with respect to understanding visual exploration as a motor action with multiple degrees of freedom.

**Data Availability Statement:** The full dataset and all analysis files are contained in a reproducible capsule on Code Ocean: https://doi.org/10.24433/CO.8767371.v2.

## Introduction

Visual exploration refers to the active process of looking around in the environment. Observers survey the environment by shifting their gaze from one location to another ("scanning") to gather visual information that supports ongoing activities [1–3]. The predominant paradigm for measuring visual exploration is recording eye movements in observers who look at screens. Although screen-based approaches yield valuable insights about how the eyes scan different types of photographs and videos, they are ill-suited for understanding visual exploration in the context of locomotion because observers must remain stationary. In contrast, mobile eye tracking studies have uncovered how gaze is adapted to different motor tasks, such as walking indoors to search an office mail room [4] or hallway [5, 6], walking outdoors over flat or uneven terrain [7–10], or even participating in an outdoor geological field expedition [11]. Yet, mobile eye tracking studies, which can measure only the position of the eyes relative to the head, miss a well-appreciated but rarely studied aspect of visual exploration. In everyday life we coordinate the rotations of the body, head, and eyes to scan in all directions [1, 12, 13].

*Gaze*—where we look in the world—is the culmination of how we rotate the eyes in relation to the head, how we rotate the head in relation to the body, and how we orient the body in space. Combining mobile eye tracking with head tracking from wearable inertial sensors [9,

**Funding:** The third author (GB) received a UC Riverside Office of Undergraduate Excellence Student Minigrant to support this project. The funders had no role in study design, data collection and analysis, decision to publish, or preparation of the manuscript.

**Competing interests:** The authors have declared that no competing interests exist.

10, 14, 15] facilitates measuring how gaze depends on nested systems—rotations of the eyes within the head are added to rotations of the head within the body. With multiple degrees of freedom to control (i.e., the eyes, head, and body), how do observers coordinate visual exploration? As we will review in the next section, the eyes and head are subject to different biomechanical constraints and have different energetic costs that shape how they are used. In spite of these constraints, the few existing studies to simultaneously measure eye and head movements suggest that there is considerable flexibility in how observers explore *within* a task [9, 16, 17]. The primary aim of the current study is to ask how exploratory eye and head movements are differentially adapted to varying demands on attention created by different tasks/ environments in the context of ongoing locomotion.

## The roles of eye and head in visual exploration

The biomechanics of eye and head movements constrain how they can be coordinated to visually explore. The oculomotor range of the eyes is ±55º along the horizontal axis [18], meaning that shifts of gaze beyond this range require the head to rotate in the same direction as the eyes. Horizontal rotations of the head in combination with eye rotations allow total gaze shifts larger than 160º. Even larger gaze shifts require the trunk to rotate and/or the feet to reorient the body in space [12, 19]. With eyes, head, and body all able to contribute to a single gaze shift, there are multiple degrees of freedom to control. For example, a 20º-amplitude gaze shift can be accomplished in many ways, even when just considering the roles of eyes and head: A 20º eye movement alone with no head movement, a 10º eye movement with a 10º head movement, or a 5º eye movement with a 15º head movement all produce the same gaze result. How, then, does the visual-motor system determine how much the eyes versus head should contribute to a gaze shift?

Laboratory studies that elicit gaze shifts to targets at different amplitudes show that the eyes alone contribute to smaller-amplitude gaze shifts (less than 20º-30º), but for larger amplitude gaze shifts the head increasingly plays a role [18, 20]. It is important to note that the head contributes to gaze shifts smaller than 55º—the limit of the eyes alone—meaning that the head is recruited even when it is not biomechanically required. This allows the eyes to stay within a more comfortable range of ±25º [21]. Although eye and head contributions appear stereotyped in laboratory tasks that simply ask participants to move the eyes to fixate a target, experimental manipulations show that they are flexibly controlled. When instructed to make two sequential gaze shifts, the head contributes more to the initial gaze shift if the second gaze shift will be in the same direction [22]. In other words, observers are more willing to rotate the head when the head will stay rotated for a while. This speaks to the different *costs* of eye versus head movements. The eyes can move quickly with little effort, whereas the head moves more slowly and requires more energy [12, 17].

The contributions of eyes and head are even more variable when measured during complex tasks. Instead of asking participants to simply fixate targets, Pelz and colleagues [16] instructed participants to copy a model, placed to the side of the participant, by arranging blocks on a workspace in front of the body. Participants turned their eyes and head to shift gaze between the model and workspace while completing the task. Unlike more controlled studies, the head contributed between 1º-10º for smaller gaze shifts (less than 15º amplitude). Most likely, participants adapted eye and head rotations from moment to moment depending on the demands of looking to the model versus workspace (and scanning back and forth between the two locations). Participants' willingness to visually explore with eyes versus head may reflect the motor costs of each movement. Indeed, a variation of block-copying task that varied the angle of the model found participants looked less frequently at the model when looking required a larger

body movement [23]. Similarly, participants comparing two similar-looking cupboards reduced the number of gaze shifts between the cupboards as the distance between the cupboards increased [17], presumably to reduce the number of costly head movements.

## How might task demands shape visual exploration with the eyes versus head?

Despite these examples of how changing the motor costs of looking (e.g., placing targets closer or farther) alters the coordination of eyes and head *within a task*, no studies have investigated how eyes versus head are coordinated to meet the informational demands *across different tasks and environments*. Mobile eye tracking studies indicate that observers tend to fixate task-relevant objects when completing tasks such as making a sandwich or cup of tea [24–27]. However, these examples—which measured eyes only—cannot reveal how both eyes and head are adapted to meet different task demands, given the flexibility and variability inherent in coordinating the eyes and head. Furthermore, locomotion—walking from one place to another—is a common "sub-task" that we must visually guide while completing a primary task, as seen in more natural tasks [28] and everyday life.

Several studies have described the role of the eyes and head in the control of walking over easy versus challenging terrain. Although these examples do not compare different task types, they demonstrate how participants adapt both eyes and head to respond to varying informational demands of locomotor control. Matthis and colleagues [10] found that in the less-demanding task of walking over flat terrain, only half of fixations were directed to the ground surface. *Spread* (or dispersion)—the standard deviation of position over the course of a task—is a commonly-used metric to examine differences in the distribution of visual exploration across tasks. 't Hart and colleagues [29] found that the horizontal spread of eye-plus-head gaze (~14º) was greater than the vertical spread of gaze (~7º), reflecting participants' propensity to visually explore targets to the left and right of the body rather than gazing down at the ground. Even though the 14º horizontal spread is well below the oculomotor range of 50º-55º, the head contributed to the horizontal spread of gaze: The horizontal spread of eye position was only 4º-5º, thus, the head accounted for the remaining portion. Similarly, Tomasi and colleagues [9] measured horizontal eye and head movements in walking participants using wearable inertial sensors, and found the head's rotation was responsible for between 37-46% of the total gaze shift amplitude across participants. Other studies of eye movements while walking over flat ground consistently find a larger horizontal than vertical spread of eye position: 14.2º versus 9.7º [30], 7º versus 5º [7], and 11.8 versus 7.2º [5].

Thus, the contributions of eyes and head during simple walking, that is, walking without a secondary task, are well characterized. Observers preferentially spread their gaze horizontally rather than vertically to visually explore the surroundings, but if walking is made more difficult the vertical spread of gaze extends down to better guide foot placement [10, 14, 29]. Moreover, the head contributes more than 35% of the rotation needed to shift gaze, even at amplitudes that are well within the limits of the oculomotor range. Our current studies build on this work to ask how eyes and head adapt to the addition of a non-locomotor task while walking, rather than altering the difficulty of walking. By adding a goal—searching for targets in a complex visual environment—we can compare the role of eye and head movements under different task demands.

How might searching while walking alter the roles of eye and head compared with walking alone? Although searching may induce participants to make larger eye movement shifts to scan more broadly within a photograph [31], this may not translate to a fully-mobile searching task. A prior study of whole-body search in virtual reality found that participants

primarily looked at mid-height regions rather than searching in areas above and below the body [32], thus, we expect search to primarily impact the horizontal component of gaze (especially with observers walking on flat ground). We predict that gaze will be spread more widely around the observer to successfully search compared to simply walking along a path. However, given the flexibility of coordinating eyes, head, and body, an increase in spread of gaze while searching could be accomplished in different ways: a larger spread of eye position without a change in head position, a larger spread of head position without a change in eye position, or increasing spread of both eyes and head. One possibility is that observers rotate the head more broadly to search in areas to the left and right of the current walking direction beyond the range of the eyes. Another possibility to rule out, however, is whether observers avoid extreme head rotations while searching if it disrupts their ability to guide locomotion. If so, we would observe an increase in the spread of eye movements but not head movements. It is important to note that we make no specific claim about the extent to which changes in the spread of eye or head movements might reflect conscious decision making. Although it is true that observers can consciously choose to employ greater head versus eye movements while exploring, it seems more likely—especially while engaged in a task like searching—that participants are not consciously deciding moment-to-moment how much to move the eyes versus head. Regardless, the current studies were not designed to distinguish between these possibilities.

## Current study

Although previous research has demonstrated the role of task in shaping eye movements, no prior work has considered how observers adapt the coordination of eye and head movements to changing task and environment demands in the context of locomotion. Whereas eye and head movements have different constraints (e.g., speed, range of movement, energetic cost), there is considerable flexibility in how much the eyes versus the head contribute to looking in different directions. We choose to compare two types of naturalistic locomotor tasks, a simple *walking* task in which participants traversed a campus path, and a *search and retrieval* task in which participants walked around a cluttered campus courtyard to find and retrieve 6 hidden targets (referred to as the *search task* for brevity). Whereas the demands on visual exploration in the walking task were minimal—participants simply needed to stay on a flat, paved path— the searching task required participants to simultaneously scan their surroundings to find targets and to start, stop, and turn while walking from one place to the next. The courtyard contained picnic tables, trees, and open concrete areas, creating a more challenging visual scene to search in as participants' view of different areas was occluded. The novel contributions of the current studies are: 1) direct comparisons of visual exploration between walking and searching tasks, and 2) comparing head movements and eye-plus-head gaze shifts, not just eye movements, across tasks.

We report two studies that employed identical procedures but differed in the data recorded. In both studies, participants' eye movements were recorded using a mobile eye tracker, and participants' walking behaviors were recorded with a GPS monitor worn on the wrist to understand the locomotor aspects of the two tasks. Study 2 added wearable inertial sensors that measured participants' head rotations. While wearing the eye tracker, GPS monitor, and (in Study 2) inertial sensors, participants completed the walking task by following a campus path from the Psychology building to an outdoor courtyard. Afterwards, participants completed the search and retrieval task in the courtyard by finding and picking up six targets (fabric squares marked with a particular shape) placed in different locations, while ignoring six distractor targets (similar looking fabric squares with a different shape).

We calculated how the *spread* of visual exploration differed between the the two tasks based on the horizontal rotation (in degrees) of the eyes (Studies 1 and 2) and head (Study 2). As in past work [5, 7, 30], spread was defined as the standard deviation of the horizontal rotation of the eyes/head and represented the degree to which participants distributed their visual exploration narrowly versus broadly over the duration of each task. As in other studies [9], we focused on horizontal eye and head movements because horizontal gaze movements are more common than vertical gaze movements when walking over flat terrain [7, 29]. Study 2 also provided an opportunity to extend laboratory studies that measured the contribution of the head to gaze shifts of varying eccentricity to a more naturalistic task. By calculating the total amplitude of each gaze shift (adding the rotations of eyes and head together), we could determine the *head contribution* (in percentage) of each gaze shift and whether that varied according to task. We predicted that the head would increasingly contribute to larger amplitude gaze shifts regardless of task, consistent with previous laboratory studies [18, 20]. Moreover, we predicted that the head would contribute more to gaze shifts in the searching task to facilitate a wider spread of gaze in the environment.

## Study 1: How are eye movements adapted to explore in different tasks/environments?

### Method

The study's procedures were designed in accordance with the Declaration of Helsinki. The UC Riverside Institutional Review Board approved the project (HS-14-137 "Eye movements during everyday activities") before data collection began. Participants gave written informed consent before the study began.

**Participants.** The final sample consisted of $N = 59$ adult participants between the ages of 18 and 31 years ($M = 20.81$ years, $SD = 3.0$, 39 female, 20 male). One additional participant was run in the study, but their data were excluded from the final sample after their eye-tracking error was found to be unusually large ($> 5º$). To be included in the study, participants were required to have normal vision or corrected-to-normal-vision with contact lenses (eye glasses could not be worn with the eye tracking headgear) and to have no motor impairments that would prevent them from engaging in the tasks. Additional participants were run in the study but excluded before data processing due to bystander interference ($n = 2$), technical difficulties (e.g., battery or SD card failure) ($n = 4$), or because the camera slipped during the searching task ($n = 7$).

Participants were undergraduate students at the University of California, Riverside who received course credit as compensation for their participation. Written informed consent was obtained at the beginning of the experimental session. Participants described their race as: White ($N = 20$), Asian ($N = 15$), Black ($N = 2$), more than one race ($N = 8$), or chose not to answer ($N = 14$). Participants described their ethnicity as: Hispanic or Latinx ($N = 27$), Not Hispanic or Latinx ($N = 29$), or chose not to answer ($N = 3$).

**Walk and search task settings.** The walking task took place along a 311-m path in the University of California, Riverside campus. Participants walked East for approximately 26 m, North for 150 m, then East for 135 m on paved sidewalks. This path took participants in between closely spaced buildings and also through a wide, open field. The walking path ended 60 m away from the courtyard, ensuring that participants could not see search target locations before they began the search task. The search arena was a courtyard that measured 45 m wide $\times$ 30 m long for a total area of 1350 m$^2$. Approximately 823 m$^2$ was garden space inaccessible to pedestrians. The remaining space was comprised of mature trees, seating areas, and wide cement walkways. Both the walking path and search arena were open to the campus

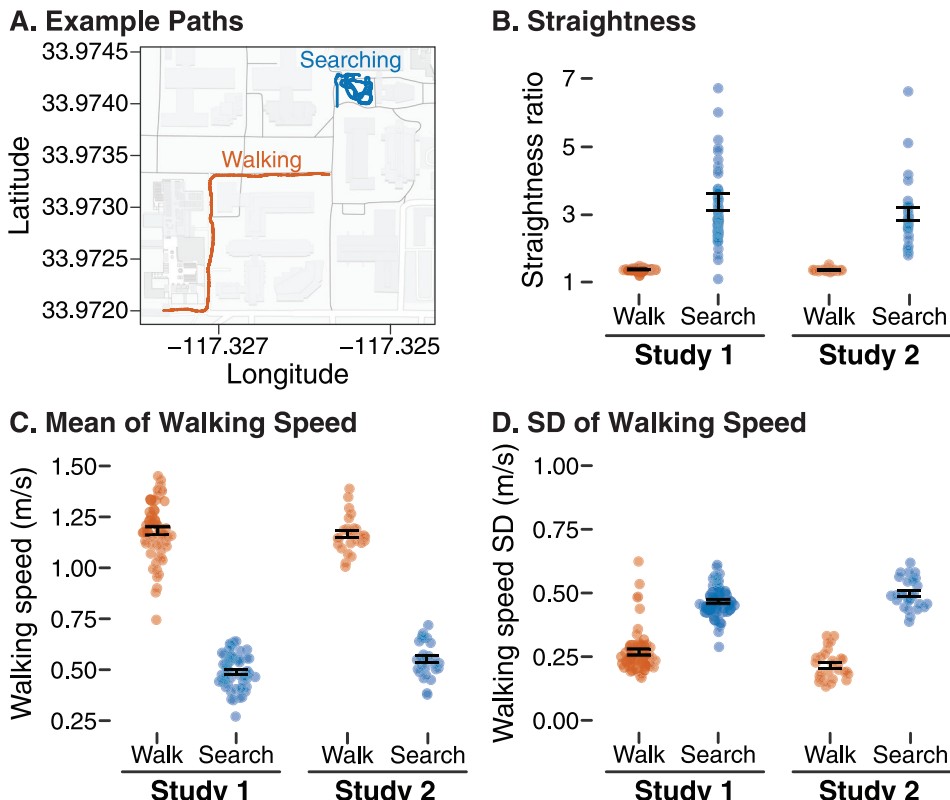

**Fig 1. Characteristics of locomotion derived from GPS data in the walking and searching tasks (orange = walking task, blue = search and retrieval task).** A) Example GPS recording of a participant's path in the walking and searching tasks overlaid on a campus map. Graphs show differences in B) straightness ratio, C) mean walking speed, and D) SD of walking speed for Studies 1 and 2 according to task. Each symbol represents a single participant's data; points are horizontally offset for visibility. Black error bars are centered on the mean and show ±1 standard error.

public, so pedestrians were often present while participants walked through both areas. Examples of one participant's GPS location overlaid on a campus map is shown for the walking and searching tasks in Fig 1A. An example video available at https://nyu.databrary.org/volume/1147 shows excerpts from the walking and searching tasks.

In the search arena, targets and distractors were fixed to trees and cement benches located throughout the courtyard in a pre-specified set of locations. Targets and distractors were 10 cm × 10 cm orange fabric squares with a 3.8 cm × 2.5 cm shape (rectangle or diamond) drawn on the front in black ink. Of the targets affixed to trees (6 total), 2 targets each were each secured 0.25 m from the ground, 1 m from the ground, and 1.5 m from the ground. Of the targets affixed to benches, targets were secured to the frame of the cement benches, never the seats or legs (each bench measured approximately 1.5 m long × 0.3 m wide).

**Eye movement and GPS recording.** A Positive Science head-mounted eye tracker was used to record the eye movements of each participant. An infrared camera that pointed towards the participant's right eye (eye camera) recorded eye movements, and the field of view of each participant was recorded by a camera that sits above the right eye and points out (field of view camera). Both eye and field of view (FOV) cameras were affixed to a modified eye glass frame that was securely hooked over each ear and held onto the participant's head with a strap. Each camera's video was fed to a recording device that was stored in a belt bag that participants wore over their right shoulder for the duration of the study. Participants wore a wide brimmed

hat to reduce eye tracker data loss from sunlight [10] and a Polar V800 Multisport GPS watch on their right wrist. The example video (https://nyu.databrary.org/volume/1147) shows real-time eye position and GPS data for an example participant.

Before the start of each task and at the end of the study, participants completed a calibration procedure that maps participant's eye position from the eye camera to their gaze location in the FOV camera. During the calibration procedure, the experimenter stood approximately 3 m from the participant and asked the participant to hold their heads as still as possible while moving only their eyes to look at locations that the experimenter indicated. The experimenter cued the participants to look at a walking stick with a brightly colored piece of cardboard at one end. The experimenter moved the colored calibration target in different locations within the FOV camera's field of view: along the central, vertical axis (top to bottom), along the horizontal axis (left to right), and along both diagonals (from corner to corner). The experimenter periodically stopped the target to allow the participant time to fixate on the calibration target without blinking or moving their head.

These video recordings were used offline (after the session) to calibrate the eye tracker using Yarbus software (Positive Science LLC), producing horizontal and vertical time series of gaze locations in field of view video (pixel) coordinates. Calibration accuracy was verified using an additional set of 5 target looks, independent from those used to calibrate the eye tracker. Calibration validation was done at the end of the walking task and at the end of the search task. For each validation point, we calculated the difference between the actual target location in the FOV camera and the gaze location in degrees—calibration error. In Study 1, participants' calibration error averaged $M = 2.73°$ ($SD = 0.69$), ranging from $1.25°$ to $3.95°$.

**Procedure.** Participants were fitted with the head-mounted eye tracker, hat, belt bag and GPS watch in the laboratory. Afterwards, the experimenter led them to a flat, shady, area outdoors for the first eye tracker calibration. The GPS watch was turned on after the calibration; this event was recorded in the eye tracker's FOV camera to allow synchronization. Next, participants completed the walking task along the prescribed path. The experimenter walked alongside the participant, providing verbal directions about where to go. At the conclusion of the walking task, the participant completed the second eye tracker calibration to account for any potential movement of the eye tracking equipment that may have occurred during the walking task.

Before the start of the search and retrieval task, the experimenter read instructions that detailed the boundaries of the search arena, explained how to identify the assigned targets versus the distractors, and how many targets were hidden (6 targets and 6 distractors). Participants were instructed to pick up each of the six targets with their hands and to leave the distractors in place. Participants were told to retrieve their targets as quickly and efficiently as possible, without running. After hearing the instructions, the search and retrieval task began. A final calibration check after the search task ensured the accuracy of the eye tracking data throughout the task.

**Data processing.** The first step in data processing was to synchronize the eye tracking and GPS time series data. The FOV camera frames that corresponded to the the GPS watch turning on/off were recorded from the FOV camera video. Using those synchronization points, we offset, scaled, and upsampled (from 1 Hz to 30 Hz) the GPS time series to match the eye tracker's time series. FOV camera videos from the eye tracker were also used to find and record the beginning and end times of each task. After synchronization, time series were extracted for horizontal eye rotation and GPS coordinates during each task to be used in subsequent analyses.

GPS coordinates were used to calculate three measures to characterize how participants walked during each task. *Walking speed* was calculated based on the length of each

participant's total walking path in each task divided by the task time. *Walking speed SD* measured the amount that participants changed their speed during each task (e.g., stopped and started walking) by calculating their instantaneous speed for each video frame, and then calculating the standard deviation of instantaneous speed across the task. Finally, the degree to which participants walked a straight path versus a circuitous path was expressed by the *straightness ratio*: the total length of the walking path divided by the shortest path between the starting and stopping points (1.0 = a perfectly straight path). Although it is expected that paths while walking will be straighter compared with paths while searching, we report these values as a way to characterize the degree of straightness to compare with future work.

Horizontal eye gaze coordinates represented how much participants rotated their eyes from left to right within the FOV camera image, measured in pixels. In order to measure eye-in-head rotations in degrees, we converted pixels to degrees based on the camera's horizontal field of view, 111º. However, the wide-angle fisheye lens meant that the pixel-to-degrees calculation could not be performed without first correcting for lens distortion [9]. We used the Matlab "Camera Calibration Toolbox" to correct the points for lens distortion before converting to degrees of visual angle. A checkerboard test image was recorded with the FOV view camera, which allowed the toolbox to create a model of the lens. The `undistortFisheye-Points` function was then used to transform each participant's raw eye movement data to remove the lens distortion. After this transformation, the eye movement data were then converted from pixels into degrees.

Using the corrected horizontal eye movement data (in degrees of rotation), we determined how much participants distributed horizontal eye movements widely versus narrowly by calculating *spread*: The standard deviation of horizontal eye position (in degrees) across each task. Fig 2A shows one participant's eye rotation distributions and corresponding spread measures in the walking and searching tasks.

## Results and discussion

Analyses were conducted in *R* [33]. Paired t-tests were used to calculate the difference in each measure between walking and searching tasks. We checked for outliers based on a threshold of 3 SD around the mean within a condition, but no outliers were found. The dataset and analysis code are shared in a reproducible "capsule" on CodeOcean (https://doi.org/10.24433/CO.8767371.v2).

**Locomotion differed between the tasks/environments.** The walking task time averaged $M = 268.1$ s ($SD = 36.5$) with participants walking a total distance of $M = 313.0$ m ($SD = 23.8$). The searching task time averaged $M = 625.4$ s ($SD = 145.7$) with participants walking a total distance of $M = 305.7$ m ($SD = 99.2$). Analysis of locomotion from GPS data illustrated the differences in behavior between the walking and searching tasks (Fig 1B–1D). In the walking task, participants' paths were straighter (straightness ratios closer to 1.0, $M = 1.38$, $SD = 0.05$), they walked more quickly (speed $M = 1.18$ m/s, $SD = 0.15$), and they walked at a more regular pace (speed SD $M = 0.28$ m/s, $SD = 0.09$). In the search and retrieval task, participants walked a more circuitous path (straightness ratio farther from 1.0, $M = 3.37$, $SD = 1.99$) at a slower average speed (speed $M = 0.49$ m/s, $SD = 0.09$), and their speed varied considerably from moment-to-moment while switching between searching for targets and stopping to retrieve them (speed SD $M = 0.47$ m/s, $SD = 0.06$). Significant paired t-tests were found comparing straightness ratios ($t(57) = -7.63$, $p < .0001$, $d = -1.00$), average walking speed ($t(57) = 33.6$, $p < .0001$, $d = 3.92$), and walking speed SDs ($t(57) = -15.0$, $p < .0001$, $d = -1.97$) between the two tasks.

**Visual exploration differed across tasks/environments.** Fig 3 (Study 1) shows that the horizontal spread of eye movements was greater in the search and retrieval task ($M = 12.9$º,

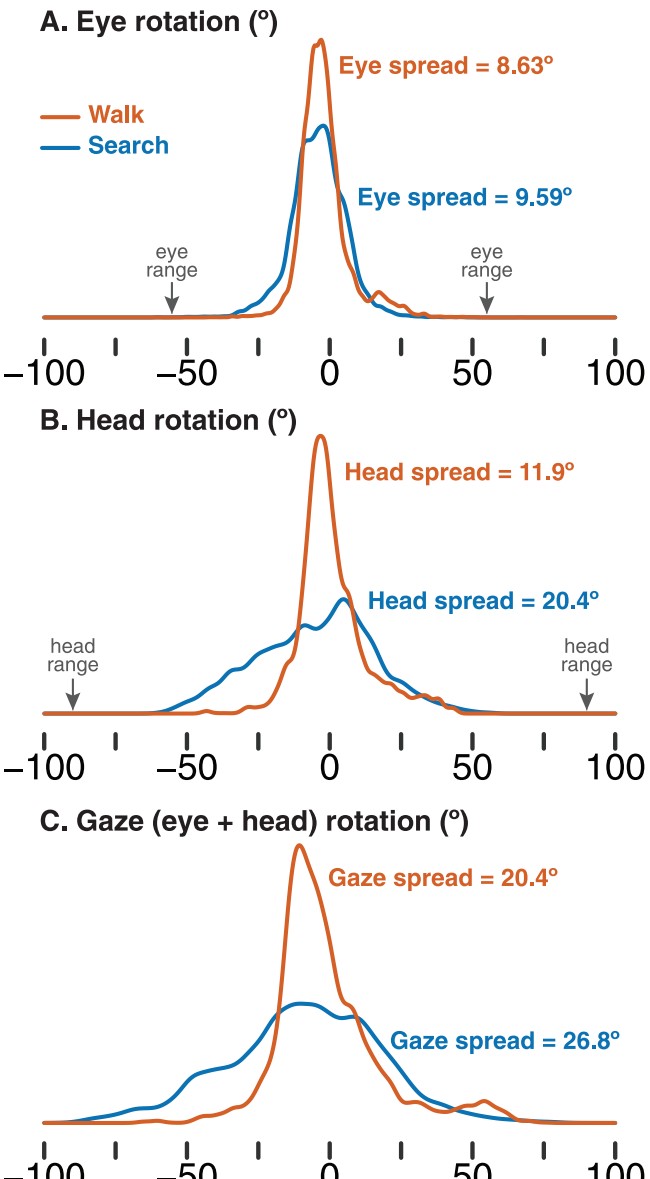

**Fig 2.** Example density plots of (A) eye rotation, (B) head rotation, and (C) gaze rotation (eyes-plus-head) for one participant. Orange lines show the distribution for the walking task and blue lines show the distribution for the search and retrieval task. Arrows indicate the approximate biological limits on (A) eye rotation and (B) head rotation for reference. Text labels show the spread of visual exploration (*SD*) based on the rotation data from each task.

*SD* = 2.00) compared with the walking task (*M* = 11.7º, *SD* = 2.59). When searching for targets, participants spent longer periods of time with their eyes rotated far to the left/right, whereas participants kept their eyes in a more narrow range within their orbits when walking without searching. This difference was confirmed by a significant paired-samples t-test between walking spread and searching spread, $t(58) = -4.18$, $p = .0001$, $d = -0.54$. Thus, participants adapted their eye movements to fit each task. With little demand on visual attention in the walking task, participants kept their eyes in a narrow window centered within the head. In contrast, participants who searched and retrieved targets broadened the scope of their eye movements to spread their gaze while looking for targets.

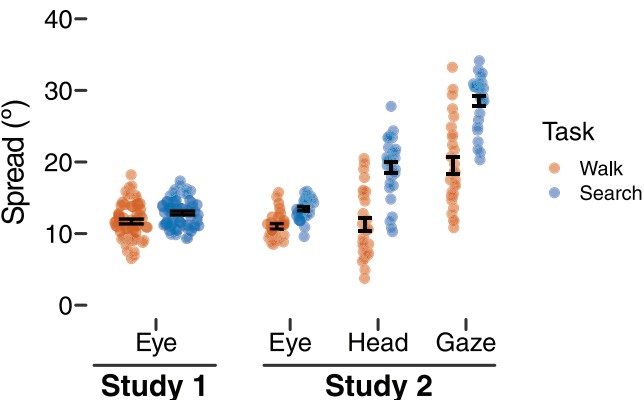

**Fig 3. Horizontal spread (standard deviation of rotational position in degrees) in the walking task (orange symbols) versus search and retrieval task (blue symbols).** Study 1 shows spread for horizontal eye movements, and Study 2 shows spread for eye movements, head movements, and gaze-in-body (eye-plus-head rotation). Each symbol represents a single participant's data; points are horizontally offset for visibility. Black error bars are centered on the mean and show ±1 standard error.

## Study 2: How are eye and head movements adapted to explore in different tasks/environments?

Study 1 indicated that participants adapted the spread of eye movements to fit the demands of the task. When walking along a straight, uniform path with no other demands on attention, participants moved their eyes within a small area. In contrast, when searching and retrieving targets participants' eyes were often rotated in different directions (within the head). However, because gaze direction in the world, relative to the body, depends on both eye and head rotation, Study 1 could not measure how much gaze was spread in different directions. It is possible that the more extreme rotations of the eyes during the search task were oppositional movements to compensate for head rotation. If so, the observer would not truly be spreading gaze more in the searching task compared with the walking task. Alternatively, if participants in the search task rotated their eyes and heads more in the same direction at the same time, then the spread of gaze when searching would truly be greater. Thus, Study 2 was designed to extend Study 1 by measuring head rotation.

### Method

The study's procedures were designed in accordance with the Declaration of Helsinki. The UC Riverside Institutional Review Board approved the project (HS-14-137 "Eye movements during everyday activities") before data collection began.

**Participants.** This study included $N = 28$ undergraduate students at the University of California, Riverside between the ages of 18 and 24 years old ($M = 20.29$ years, $SD = 1.43$, 16 male, 12 female). Participants were recruited from the psychology department participant pool at the University of California, Riverside and received course credit as compensation for their participation in this study. To be included in the study, participants needed to have normal or corrected-to-normal vision without wearing eyeglasses and were required to have no motor impairments that would prevent them from engaging in the tasks. Each participant gave informed consent at the beginning of the experimental session. Participants described their race as: Asian ($N = 13$), White ($N = 7$), Black ($N = 2$), Native Hawaiian or other pacific islander ($N = 1$), more than one race ($N = 1$), or chose not to answer ($N = 4$). Participants described their ethnicity as: Hispanic or Latinx ($N = 8$) or Not Hispanic or Latinx ($N = 20$). Five

additional participants completed the study, but their data were ultimately excluded from the final sample due to technical difficulties ($n = 3$), the camera slipping out of place during the searching task ($n = 1$), and bystander interference ($n = 1$).

As in Study 1, calibration validation was performed at the end of the walking task and at the end of the search task. Calibration errors for the 28 participants averaged $M = 3.50º$ ($SD = 0.81$), ranging from $1.70º$ to $4.57º$.

**Head movement recording.**   All procedural aspects of Study 2 were equivalent to Study 1, with the only change being the addition of wearable inertial motion sensors that recorded head position. Two STT systems (STT-IWS) motion sensors were worn throughout the duration of the entire study. One sensor was placed on the seventh cervical vertebra (C7) using a Velcro chest harness and the other was secured on top of the participant's head (underneath the wide-brimmed hat) with a Velcro headband. To facilitate synchronization of the motion sensors with the eye tracking data, participants were instructed before each eye tracking calibration to hold their heads still and look straight ahead and then to make a quick head rotation to the left and then to the right.

**Data processing.**   Measures of walking from GPS data and measures of eye movement spread were processed as in Study 1. To integrate head rotation measures with eye movement and GPS data, we extracted head rotation time series from the STT systems using their proprietary iSen software. The software calculated time series of head position (400 Hz) from the acceleration and gyroscope data collected by the head sensor, using the C7 sensor as a reference point. To synchronize the head movement time series to the eye-tracking time series, we identified the head-turn synchronization events in the eye tracker's FOV camera video (moment that the field of view changed during the rapid head rotation) and the matching timestamp from the head rotation time series data from a plot. Based on the synchronization event times at the beginning and end of the session, we offset, scaled, and downsampled the head rotation data to match the eye movement and GPS time series. Eye movement data were undistorted and converted into degrees as in Study 1, resulting in synchronized time series of horizontal eye and head rotation in the same measurement units. The example video (https://nyu.databrary.org/volume/1147) shows head rotation data synchronized with eye rotation and GPS.

Head rotation spread was calculated in the same way as eye movement spread. In addition, we calculated a gaze-in-body time series by adding eye and head rotations together (negative rotations corresponding to left, 0 corresponding to center, and positive rotations corresponding to right). We calculated gaze-in-body spread from this time series (the standard deviation of gaze position) to determine the overall distribution of gaze relative to the observer's body. Fig 2B and 2C shows one participant's head and gaze rotation distributions and corresponding spread measures in the walking and searching tasks.

## Results and discussion

We compared locomotion (straightness, walking speed, and walking speed SDs), visual exploration (spread of eye/head movements), and the head contribution to gaze shifts across tasks. With the additional factor of eye versus head movements, we employed linear mixed-effect models (LMMs) in R using the *lme4* package [34] with participant as a random effect. Maximal models that included random slopes of fixed factors failed to converge, so only random intercepts of participant were included. Significance tests for LMMs were calculated using the *lmerTest* package [35] implementation of the Satterthwaite correction. Pairwise follow-up tests were corrected for multiple comparisons using the Holm-Bonferroni correction. All measures were checked for outliers according to a 3-*SD* criterion, but none were found. The data and

analysis code are available in the same CodeOcean capsule as Study 1 (https://doi.org/10.24433/CO.8767371.v2).

**Locomotion differed across tasks/environments.** The walking task time averaged $M = 279.7$ s ($SD = 20.5$) with participants walking a total distance of $M = 324.2$ m ($SD = 12.0$). The searching task time averaged $M = 731.1$ s ($SD = 157.0$) with participants walking a total distance of $M = 407.6$ m ($SD = 125.2$). The three GPS-derived measures of locomotion differed according to task, mirroring the results of Study 1 (Fig 1B–1D). When completing the walking task, paths were straighter ($M = 1.37$, $SD = 0.04$), walking speed was greater ($M = 1.16$ m/s, $SD = 0.09$), and they walked at a more regular pace (speed SD $M = 0.22$ m/s, $SD = 0.06$). When searching, paths were less straight ($M = 3.02$, $SD = 1.01$), average walking speeds were slower ($M = 0.55$ m/s, $SD = 0.09$), and speed varied more (speed SD $M = 0.50$ m/s, $SD = 0.06$). Significant paired t-tests were found comparing straightness ratios ($t(27) = -8.57$, $p <.0001$, $d = -1.62$), average walking speed ($t(27) = 24.3$, $p <.0001$, $d = 4.6$), and walking speed SDs ($t(27) = -18.6$, $p <.0001$, $d = -3.51$) between the two tasks.

**Visual exploration differed across tasks/environments.** Fig 3 shows the spread of visual exploration for the eyes, head, and gaze (eyes-plus-head) for Study 2. Consistent with our prediction, gaze was spread more broadly during the search task ($M = 28.5º$, $SD = 3.65$) compared with the walking task ($M = 19.5º$, $SD = 6.01$; $t(27) = -8.82$, $p <.0001$, $d = -1.67$).

How were eyes and head adapted between the walking and searching tasks to spread gaze-in-body more broadly when walking and searching? We used a 2 task (walking vs searching) × 2 effector (eyes vs head) LMM to model spread based on task and effector as fixed factors and participant as a random intercept. Replicating Study 1, and consistent with the gaze result in the previous paragraph, a significant main effect of task, $F(1, 81) = 80.18$, $p <.0001$, indicated that spread was greater when searching compared with walking. A significant main effect of effector, $F(1, 81) = 26.68$, $p <.0001$, and a significant task × effector interaction, $F(1, 81) = 22.70$, $p <.0001$, reveal that the increase in gaze spread from walking to searching was more dependent on the head compared with the eyes. When walking, the spread in head position ($M = 11.3º$, $SD = 4.72$) and eye position ($M = 11.1º$, $SD = 1.94$) were similar, and spread did not significantly differ in a pairwise comparison between eyes and head ($p = .77$). In contrast, head position spread in the searching task ($M = 19.2º$, $SD = 4.30$) was significantly greater than the spread in eye position ($M = 13.5º$, $SD = 1.5$; $p <.0001$).

Thus, the spread of both eye and head movements increased from walking to searching, allowing gaze to be distributed more broadly in the environment when looking for and retrieving hidden targets. However, the adaptation of spread was more pronounced in head movements compared with eye movements.

**Head contribution to gaze shifts differed across tasks/environments.** The final set of analyses examined the *head contribution* to gaze shifts to different eccentricities relative to the body in the two tasks. Using the gaze-in-body time series, we identified local minima (shifts to the left of the body) and maxima (shifts to the right of the body) using Matlab's *findpeaks* function. Peaks were required to be a minimum of 10 video frames (333 ms) apart and were only recorded during times that both eyes and head were rotated in the same direction. For each peak, we calculated the head contribution as the percentage of the gaze shift accomplished by the head. For example, if the eyes rotated 20º to the left and the head rotated 20º to the left for a combined eccentricity of 40º, the head contribution would be half (50%) of the total eccentricity. Fig 4 shows three examples of the head's contribution to gaze shifts of different eccentricities (the black arrow indicates the total eccentricity of the shift, the green shaded region indicates the amount the head rotated, and the gray region represents the additional rotations of the eyes). In order to analyze the relative contribution of the head as a function of the total eccentricity of the gaze shift, we found each participant's average head contribution by

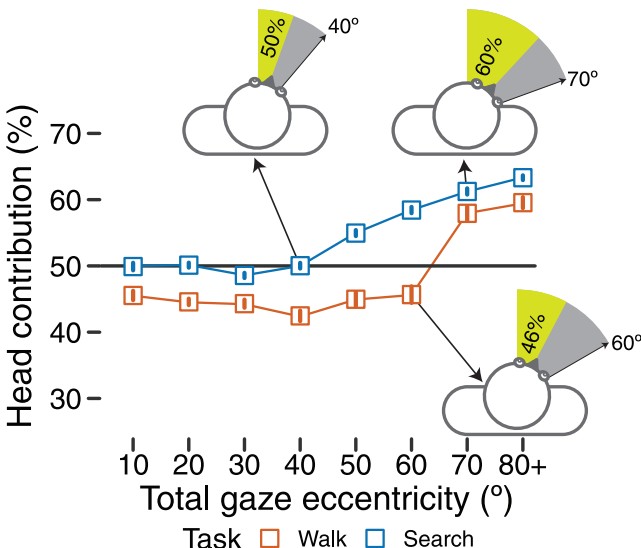

**Fig 4. Head contribution to gaze shifts of varying eccentricity (x-axis).** Each symbol shows the mean head contribution to a gaze shift—the percentage of the gaze shift accomplished through head rotation as opposed to eye rotation. Symbols above the black horizontal line at 50% indicate that the head contributed more than the eyes; symbols below 50% indicate that the eyes contributed more to the gaze shift compared to the head. Orange symbols represent the walking task and blue symbols represent the search and retrieval task. Error bars (within the symbols) indicate ± 1 standard error. Three top-down drawings of an observer depict the eye contribution (gray shading) versus head contribution (green shading) for gaze shifts at 40º in the search task, 60º in the walking task, and 70º in the search task.

aggregating over peaks in eight 10º-wide bins (i.e., total shifts 10º-20º, 20º-30º, 30º-40º, 40º-50º, 50º-60º, 60º-70º, 70º-80º, and 80º+). In Fig 4, each bin is labelled by the lower bound of the bin (e.g., 10º-20º is labelled 10º).

Fig 4 shows that more eccentric gaze shifts recruited a greater head contribution in both walking and searching tasks, suggesting that previous laboratory results [18, 20] generalize to a naturalistic locomotor task. Unlike laboratory tasks, the head contributed to even the smallest shifts of gaze (10º). Visual inspection of Fig 4 suggests that eyes and head played consistent, near-equal roles at smaller eccentricities (less than 50º), but the head increasingly contributed at larger eccentricities. However, the head contributed more in the searching task compared with the walking task at every eccentricity. These results were confirmed by a 2 task × 8 eccentricity LMM on head contribution with random intercepts by participant, which revealed a significant main effect of task, $F(1, 425.12) = 93.58$, $p < .0001$, and a significant main effect of bin, $F(7, 424.97) = 8.02$, $p < .0001$. Although it appeared that the increase in head contribution occurred at different eccentricities for each task (between 40º-50º for the searching task but between 60º-70º for the walking task), the task × eccentricity interaction was non-significant ($p = .14$). Pairwise comparisons between tasks at every eccentricity were statistically significant ($ps < .047$), confirming that the head contributed more when searching regardless of the eccentricity of the gaze shift.

## General discussion

To summarize, the current study investigated how task and environment affect the spread of eye and head visual exploration during outdoor locomotion. We found that that eye and head movements are adapted differently when walking along a path (walking task) compared with walking around a cluttered courtyard while searching for and retrieving targets (searching

task). More specifically, individuals spread their gaze (relative to the body) more broadly during the search and retrieval task compared to the walking task through a large increase in the spread of head movements paired with a modest increase in the spread of eye movements. We also extended a laboratory effect—that the head contribution to a gaze shift increases as a function of the amplitude of a gaze shift—to show that it holds in walking observers, and additionally showed that the degree of head contribution changes depending on the task/environment. The head's contribution to gaze shifts was greater while searching compared to when walking for gaze shifts of every amplitude.

There is abundant research from both screen-based [31, 36–38] and mobile eye tracking studies [16, 24, 26] showing that eye gaze is adapted to the observer's task. As expected, we found in Study 1 that the spread of eye movements increased modestly when searching compared with walking (12.9º versus 11.7º). Given that the horizontal eye spread in previous walking studies ranged from 5º-14º [5, 7, 29, 30], a task difference of 1.2º appears quite small, even though it was statistically significant. Yet, measuring the eyes alone tells only part of the story. As expected, the degree to which gaze-in-body changed between tasks was large (28.5º for searching versus 19.5º for walking in Study 2), demonstrating that the two tasks placed very different demands on visual exploration that were not apparent from examining the movements of the eyes alone. Indeed, the largest adaption was evident in movements of the head, with a spread of 19.2º in head position observed while searching compared to only 11.3º while walking. The differential contributions of eyes and head show the value of measuring head position during visual exploration. Research using eyes-only measures of visual exploration should be especially cautious in the treatment of null effects if the head's contribution is not characterized.

Given the winding, circuitous paths participants took through the courtyard when searching (Fig 1), it was expected that participants would distribute gaze more broadly around the environment to explore while searching. However, the flexibility in how the eyes, head, and body can contribute to gaze shifts means that the eyes alone, the head alone, or eyes and head in different combinations could have been adapted to meet the demands of the searching task. Indeed, the gaze density plot in Fig 2 (bottom) shows that most shifts of gaze were well within the biomechanical range of the eyes and head. But despite the multiple degrees of freedom afforded to participants, they arrived at a similar solution: increasing the spread of *both* eyes and head when searching, but increasing the spread of the head by a greater degree. Whether this is the most optimal or efficient strategy remains to be tested. Indeed, we cannot claim from the present work that energetic cost is the critical factor in shaping how eyes versus head contribute. Although head movements are more energetically costly, they also generate vestibular and proprioceptive information that eye movements do not. Future work could experimentally restrict head movement or increase the energetic cost of head movements to determine: 1) whether the eyes compensate by increasing their spread when head movement is reduced, and 2) whether a diminished contribution of the head to visual exploration degrades search performance.

Finally, measuring concurrent eye and head movements afforded us an opportunity to ask how the eyes and head contribute to gaze shifts of varying amplitude. Whereas the comparisons of head versus eye speed/spread were temporally coarse (aggregating across the entire task), measuring the the eye and head contributions to each gaze shift showed how they were coordinated in the moment. Like Tomasi and colleagues [9], who studied eye and head rotations in natural outdoor locomotion, we replicated the laboratory finding that the head contribution to gaze shifts increases as the total amplitude increases [18, 20, 21]. Our investigation extends those prior studies to show that this is true both while walking and searching in more naturalistic situations. Moreover, our study adds a novel finding: The relative contributions of

eyes and head change as a function of task/environment, not merely amplitude, as evidenced by an overall greater head contribution in the search task. This suggests that the overall strategy of visual exploration changed in the searching task—the head was not just recruited to look at extreme locations, but contributed more to visual exploration in all locations. Perhaps, the head contributed more to smaller shifts of gaze in the searching task in anticipation of subsequent, larger shifts in the same direction, as in previous laboratory work [22]. How much this strategy is a conscious choice of the participant remains to be tested. Although participants might introspectively recognize that they "look around" more in the searching task, it seems unlikely that they are aware of precisely how much they adapted movements of the eyes versus head. Since visual exploration is over-learned—we continually shift gaze from moment to moment—observers may automatically adjust their exploration to suit the task. Developmental studies of visual exploration in infants and children may shed light on how exploratory control is acquired.

We acknowledge several limitations in our study that can be addressed in future research. First, we designed the study to use two different environments, each paired with a different task, to create unique demands on visual attention. Although this was helpful for using locations that fit with each activity (e.g., the walking path did not contain locations that would have been suitable for hiding targets), it also makes it more difficult to interpret what differences between the conditions were most important for changing visual exploration. In future work, we can compare walking with walking and searching in the same environments to better tease apart how the demands of the task and the visual features of the environment may have contributed to visual exploration. We also note that aggregating visual exploration across the entire walking task and entire searching task is an oversimplification. Although it was a useful way to broadly characterize how the spread of visual exploration differs across the two tasks, we are unable to address how moment-to-moment changes in actions and goals within each task (i.e., searching, retrieving, navigating during the search task) may have changed visual exploration over time. Finally, we acknowledge that the current studies cannot address the degree to which the selection of eye and head movements reflect conscious versus automatic processes.

In conclusion, the current studies show the importance of measuring both eyes and head to understand gaze behavior in complex, real-life tasks. Although differences were apparent in eye movements alone (Study 1), studying eye and head movements together uncovered that each effector contributed differently to visual exploration (Study 2). Adaptations to eyes-plus-head gaze were evident both in aggregate across the task as well as at the level of moment-to-moment gaze shifts, showing that the entire visual exploratory system was adapted to meet task demands. Our study shows the feasibility of using wearable, wireless eye and head tracking to characterize behavior "in the wild"; this method can be used profitably to investigate eye-head adaptation in a wider range of tasks across different environments. In doing so, we may better understand how visual exploration meets the various demands of daily life.

## Acknowledgments

The authors are grateful to Adonis Salazar, Cruz Hernandez, Jaspreet Kaur, Stephanie Martinez, Jennifer Escobar, and the other members of the Perception Action and Development Lab for their work on this project.

## Author Contributions

**Conceptualization:** John M. Franchak, Brianna McGee, Gabrielle Blanch.

**Formal analysis:** John M. Franchak.

**Funding acquisition:** Gabrielle Blanch.

**Investigation:** Brianna McGee, Gabrielle Blanch.

**Methodology:** John M. Franchak, Brianna McGee, Gabrielle Blanch.

**Project administration:** John M. Franchak, Brianna McGee, Gabrielle Blanch.

**Resources:** John M. Franchak.

**Software:** John M. Franchak.

**Supervision:** John M. Franchak.

**Validation:** Brianna McGee, Gabrielle Blanch.

**Visualization:** John M. Franchak.

**Writing – original draft:** John M. Franchak, Brianna McGee, Gabrielle Blanch.

**Writing – review & editing:** John M. Franchak, Brianna McGee, Gabrielle Blanch.

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
