## [Decision Letter · Decision Letter 0]

27 Jun 2021

PONE-D-21-17566

Adapting the coordination of eyes and head for task-specific visual exploration in the context of locomotion

PLOS ONE

Dear Dr. Franchak,

Thank you for submitting your manuscript to PLOS ONE. After careful consideration, we feel that it has merit but does not fully meet PLOS ONE’s publication criteria as it currently stands. Therefore, we invite you to submit a revised version of the manuscript that addresses the points raised during the review process.

The Reviewers have made several suggestions for clarification, and improved presentation. Please respond to these comments, making any changes that are appropriate and explaining your decisions.

We look forward to receiving your revised manuscript.

Kind regards,

Thomas A Stoffregen, PhD

Academic Editor

PLOS ONE

Journal Requirements:

Reviewers' comments:

Reviewer's Responses to Questions

**Comments to the Author**

1. Is the manuscript technically sound, and do the data support the conclusions?

Reviewer #1: Yes

Reviewer #2: Yes

Reviewer #3: Yes

2. Has the statistical analysis been performed appropriately and rigorously? 

Reviewer #1: Yes

Reviewer #2: Yes

Reviewer #3: Yes

3. Have the authors made all data underlying the findings in their manuscript fully available?

Reviewer #1: Yes

Reviewer #2: Yes

Reviewer #3: Yes

4. Is the manuscript presented in an intelligible fashion and written in standard English?

Reviewer #1: Yes

Reviewer #2: Yes

Reviewer #3: Yes

5. Review Comments to the Author

Reviewer #1: The manuscript “Adapting the coordination of eyes and head for task-specific visual exploration in the context of locomotion” explored the coordination between eye and head movements when walking along a straight path compared with walking around circuitous path while searching for targets. The authors found that the degree of head’s contribution to gaze shifts was greater during the search and retrieval task compared to the walking task, which led to the overall increase in the spread of visual exploration in the search task. For the most part, the study described in this paper was well-motivated and competently conducted. However, I think that minor revisions are required before I could recommend publication.

Point 1.

As the authors described in the text, the two experimental conditions differed not only in the task performed (walk versus search) but also in the environment in which the task was performed (straight versus winding, circuitous path). This should be made explicit in the title of the manuscript, the subtitles of Study 1 and 2 and elsewhere, because the gaze measures might have reflected the difference in the environment, in addition to the task difference.

Point 2.

p. 10, line 260: “The walking task took approximately 5 minutes to complete.” - The authors reported mean speed was 1.16 m/s and the length of the path was 211 m, according to which the walking task should have taken about 3 minutes to complete on average. I wonder why these values are discrepant.

Point 3.

Related to Point 2, readers would find it helpful if the authors could provide mean and SD of task duration in walking condition, as well as mean and SD of the distance traveled in search condition.

Point 4.

The authors computed the horizontal spread of eye movements and head movements at a single timescale (SD computed over the task duration). I wonder if the information about how the amount of spread increases as a function of the length of time window (i.e., Hurst exponent) may further reveal the difference between the conditions (c.f., Viswanathan et al., 2011, The Physics of Foraging). This is just a comment, not a request.

Reviewer #2: Abstract

- No comments

Introduction

- Please clarify the following sentence: “In other words, observers are more willing to rotate the head when it is going to stay a while.” Specifically clarify the “it” that the authors refer to.

- There is a typo on line 89.

- The authors allude to observers “selecting” how much to move the eyes versus the head. Please state whether you believe this to be a conscious cognitive “decision” or if a different conceptual argument is being made.

Method

- I presume that the participants had no motor impairments, but this is not explicitly stated.

- Please provide a rationale for why the tracker only focused on the right eye. Is there an argument for a “dominant” eye is participants? If so, was everyone right-handed and right eye dominant?

Results

- No comments

Discussion

- Please provide a theoretical explanation for your findings (related to the last comment about the introduction).

Reviewer #3: General impression: This is a well-written manuscript describing a study that aimed to understand the interplay between the eyes and the head in visual exploration while walking and how the dynamics may alter as a function of task demands. This is an interesting yet important subject as it paves the road for future elaboration on other sensory systems that may be involved in and supportive of the process of “visual exploration.”

Comments:

• Page 2, line 17: “…, the eyes within the body,…” It would make more sense if it is stated as “the head in relation to the body”.

• Page 4, line 89-100: I was confused by the semantic use of “spread”; I thought the authors meant to say the range of movement. I can imagine the word “spread” being used by prior related literature and consequently the authors may choose to stick with the convention. Please consider including a brief clarifying statement to alert the readers about the operational use of the word – “spread” – before waiting until the methods section (e.g., page 7, line 162-165).

• Page 11, line 288-291: About straightness ratio, I am not certain about the relevance of this parameter being included in the study. Given the walking task and the search task have very different goals and environmental settings, I would be surprised if the there is no difference in straightness ratio between the two tasks. Perhaps, the authors could include some brief explanation to help the audience better understand the important of this parameter.

• Discussion: The authors, probably influenced by previous literature on visual exploration, seemed to suggest the utilization of head movement even when the task could be achieved by eye movement only may be due to energetic cost. While the authors also presented this unresolved issue as a study limitation. I suggest the authors to consider the other sensory systems (e.g., vestibular, neck proprioception) that may be involved during head movement as to potentially explain why the involvement of head movement may be beneficial while performing these tasks.

• Figure 1 and Figure 2: These two figures are incorrectly placed and referred.

• References: Please check your citation format. Some are incorrect.

6. PLOS authors have the option to publish the peer review history of their article (what does this mean?). If published, this will include your full peer review and any attached files.

Reviewer #1: No

Reviewer #2: No

Reviewer #3: No

---

## [Author Response · Author response to Decision Letter 0]

13 Jul 2021

Please see the attached "Response to Reviews" document.

---

## [Decision Letter · Decision Letter 1]

9 Aug 2021

Adapting the coordination of eyes and head to differences in task and environment during fully-mobile visual exploration

PONE-D-21-17566R1

Dear Dr. Franchak,

We’re pleased to inform you that your manuscript has been judged scientifically suitable for publication and will be formally accepted for publication once it meets all outstanding technical requirements.

Kind regards,

Thomas A Stoffregen, PhD

Academic Editor

PLOS ONE

Additional Editor Comments (optional):

Reviewers' comments:

Reviewer's Responses to Questions

**Comments to the Author**

1. If the authors have adequately addressed your comments raised in a previous round of review and you feel that this manuscript is now acceptable for publication, you may indicate that here to bypass the “Comments to the Author” section, enter your conflict of interest statement in the “Confidential to Editor” section, and submit your "Accept" recommendation.

Reviewer #1: All comments have been addressed

Reviewer #2: All comments have been addressed

Reviewer #3: All comments have been addressed

2. Is the manuscript technically sound, and do the data support the conclusions?

Reviewer #1: Yes

Reviewer #2: Yes

Reviewer #3: Yes

3. Has the statistical analysis been performed appropriately and rigorously? 

Reviewer #1: Yes

Reviewer #2: Yes

Reviewer #3: Yes

4. Have the authors made all data underlying the findings in their manuscript fully available?

Reviewer #1: Yes

Reviewer #2: Yes

Reviewer #3: Yes

5. Is the manuscript presented in an intelligible fashion and written in standard English?

Reviewer #1: Yes

Reviewer #2: Yes

Reviewer #3: Yes

6. Review Comments to the Author

Reviewer #1: I think that the authors have adequately addressed my concerns. The quality of the manuscript is improved.

Reviewer #2: The authors have satisfactorily responded to the reviewers' comments. I believe that the manuscript is now ready for publication.

Reviewer #3: (No Response)

7. PLOS authors have the option to publish the peer review history of their article (what does this mean?). If published, this will include your full peer review and any attached files.

Reviewer #1: No

Reviewer #2: No

Reviewer #3: No

---

## [Editor Report · Acceptance letter]

12 Aug 2021

PONE-D-21-17566R1 

Adapting the coordination of eyes and head to differences in task and environment during fully-mobile visual exploration  

Dear Dr. Franchak:

I'm pleased to inform you that your manuscript has been deemed suitable for publication in PLOS ONE. Congratulations! Your manuscript is now with our production department. 

Kind regards, 

on behalf of

Dr. Thomas A Stoffregen 

Academic Editor

PLOS ONE